# Implementation and Evaluation of 5G MEC-Enabled Smart Factory

Nadhif Muhammad Rekoputra [1], Chia-Wei Tseng [1], Jui-Tang Wang [1], Shu-Hao Liang [2,*], Ray-Guang Cheng [1,*], Yueh-Feng Li [3] and Wen-Hao Yang [3]

1    Department of Electronic and Computer Engineering, National Taiwan University of Science and Technology, Taipei City 106335, Taiwan
2    Graduate Institute of Intelligent Manufacturing Technology, National Taiwan University of Science and Technology, Taipei City 106335, Taiwan
3    Wireless Communication Laboratory, Telecommunication Laboratory of Chunghwa Telecom, Taipei City 106335, Taiwan
*    Correspondence: shuhaoliang@mail.ntust.edu.tw (S.-H.L.); crg@mail.ntust.edu.tw (R.-G.C.); Tel.: +886-2-2733-3141 (ext. 5207) (S.-H.L.)

**Abstract:** A 5G network can provide more comprehensive bandwidth connectivity for the industry 4.0 environment, which requires faster and tremendous data transmission. This study demonstrates the 5G network performance evaluation with MEC, without MEC, WiFi 6, and Ethernet networks. Usually, a 5G network engages with Multi-access Edge Computing, providing the computing functions dedicated to the users on edge nodes. The MEC network architecture presents significant facilities, a network schematic, and data transmission routers. The field test performs high-definition streaming video and heavy-traffic load testing to evaluate the performance based on different protocols by comparing throughput, latency, jitter, and packet loss rate. MEC network performance, streaming video performance, and load test evaluation results reveal that the 5G network working with MEC achieved better performance than when it was working without MEC. The MEC can improve data transmission efficiency by dedicated configuration but is only accessible with authentication from mobile network operators (MNOs). Therefore, MNOs should offer industrial private network users partial authentication for accessing MEC functionality to improve network feasibility and efficiency. In conclusion, this work illustrates the 5G network implementation and performance measurement for constructing a smart factory.

**Keywords:** 5G network; Multi-access Edge Computing (MEC); WiFi 6; industry 4.0





## 1. Introduction

5G technology innovation inspires the growth of global mobile network subscribers. The subscription could increase from 236 million in 2020 to 4800 million in 2026, based on the report from Statista in 2022 [1]. Device manufacturers, business intelligence software firms, mobile carriers, system integrators, and infrastructure vendors can all play unique but complementary roles across the IoT and other services landscape [2]. The application of the 5G network is quite extensive, and industry 4.0 is one of the essential application scenarios. 5G network implementation at factories encounters more challenges than the 5G application for consumers since the industries continue pushing the boundaries of wireless technologies in workplaces. Many 5G subscriptions can be from industrial users—machine type connection. On the other hand, time-critical communication (TCC) and massive machine-type communication (mMTC) has become the new challenge of 5G network implementation in industry 4.0.

In industrial applications, mobile machines or vehicles require a wireless connection to the applications on cloud services. The 4G LTE generation, the predecessor of 5G, already provided Mobile Cloud Computing (MCC) [3] that allows connected devices to access

remote cloud data centers through mobile networks to run computing or data-intensive service to improve performance and extend the battery life of the end device. However, those connected devices would take hundreds of kilometers of long-distance transmission to access the remote cloud center with MCC, resulting in high latency and increasing data loss. Satyanarayanan et al. (2015) also depict an open ecosystem for mobile cloud convergence and mention the relative MCC issues [4].

Most industry executives believe the 5G network can provide the solution for applications regarding ultra-reliable low latency communication (URLLC) and massive machine-type connection (mMTC). Nevertheless, 5G network architecture requirements can differ between industries such as electronica, steel, chemical, automobile, and machinery, and there are no templates or references to follow or duplicate. Deploying 5G networks requires evaluation processes or simulation to prove the design or concepts. Ramiro et al. (2022) introduced the 5G network enabling technology that reveals the 5G network planning process and introduces the software tools for monitoring and emulating networks, which can be an example of 5G network planning [5].

Multi-Access Edge Computing (MEC) [6] enhances edge capability through computational and storage resources called edge servers. A packet from clients is sent to the mobile edge but not passed through the core networks as the cloud architecture had been presented by Nikaein et al. [7] and Huang et al. [8]. This approach can potentially improve communication efficiency and shorten the interval time in data transmission.

Figure 1 shows the initiative 5G network with a MEC server and peripherals like edge server, switch, and base station. The MEC server provides computation, storage, and network resources closer to user equipment (UE), which can lower data transmission latency.

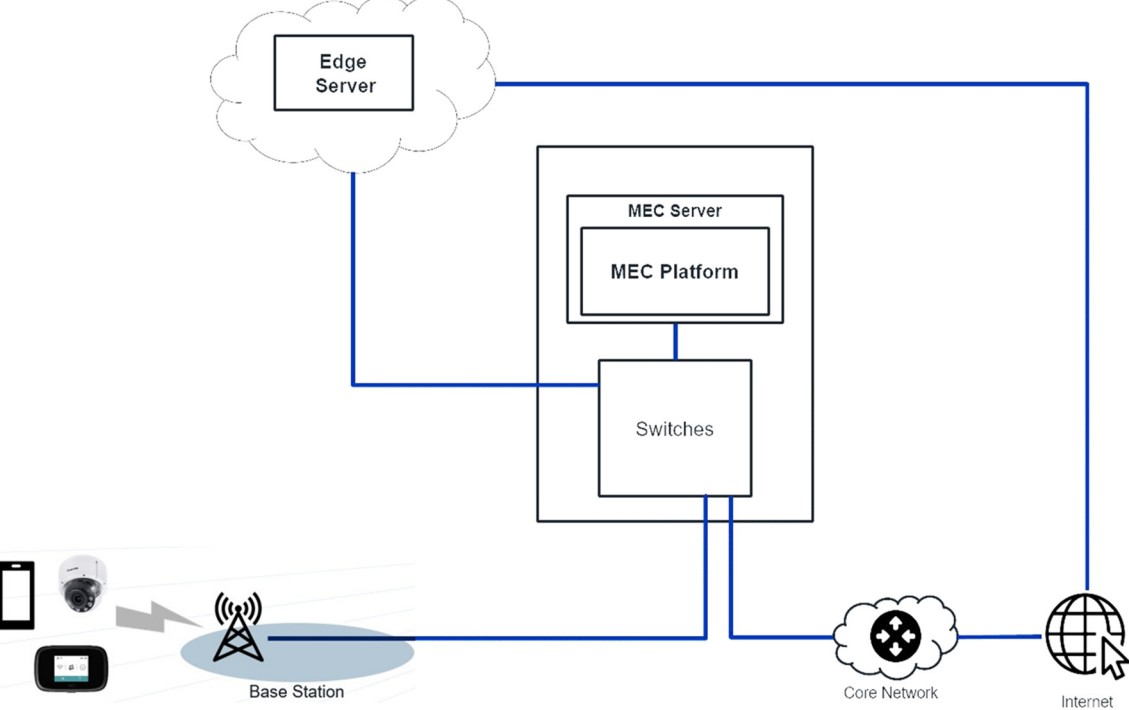

**Figure 1.** UE and Multi-access Edge Computing (MEC) initiative.

The performance of MEC presents task offloading capability and task scheduling efficiency to realize the goal of a low-latency service, referring to the works by Chen et al. (2018) [9] and Liu et al. (2016) [10], respectively. Further, an exercise combines task offloading and computation optimization in a practice of MEC by Wang et al. (2018) [11]. Moreover, Quadri et al. (2018) [12] conducted relevant work on storage performance and quality of service measurement.

Moreover, a state-of-the-art 5G network simulation for the performance evaluation of MEC was performed by Virdis A. et al. (2020) [13]. The work conducted network simulations on the Simu5G simulator and Intel CoFluent Studio. The Simu5G simulates the 5G NR communication, while the Intel CoFluent Studio models optimize the MEC and UPF environment.

In this work, the 5G MEC network performance evaluation utilized actual physical equipment operated by Mobile Network Operator (Chunghwa Telecom, Taiwan). Table 1 shows the 5G MEC models, the state-of-the-art 5G network simulation, and the experiment model at NTUST.

**Table 1.** 5G MEC Model—simulation model (Virdis) and experiment model (NTUST).

| Item | Model | |
|------|------------------------|-----------------------------|
| | **Virdis A. et al. (2020) [13]** | **5G MEC test@NTUST** |
| 5G NR | Simu5G Simulation | 5G Smallcell + 4G/5G Base Station |
| 5G CN (UPF) | Intel CoFluent Studio | CHT Core Network |
| MEC Platform | Intel CoFluent Studio | CHT MEC Server |
| MEC APP | Intel CoFluent Studio | AAEON Server |

Even though the Multi-Access Edge Computing research activity has increased in the past few years, this field still lacks a systematic study on the performance comparison of the Multi-Access Edge Computing network with real devices and different access technologies. That motivates us to construct a testbed of MEC at the machining workshop of the Industry 4.0 Implement Center at the National Taiwan University of Science and Technology (NTUST). The prospective contribution delivered in this work is as follows:

- A field test performance evaluation;

The 5G MEC network conducts the data transmission to evaluate the throughput, latency, jitter, and packet loss rate with the Ethernet, WiFi 6 mesh (802.11ax) network as a paradigm for relevant experiments.

- Streaming video transmission speed assessment;

This practice depicts the video streaming transmission rate with different network configurations or services, including 5G with MEC, 5G without MEC, and WiFi 6 mesh networks.

- Load testing on the MEC network;

The MEC server runs the MEC Platform, provides system information, including CPU and memory usage, and shows massive traffic and user records. Load testing applies on the MEC server to seed the overall network's performance.

- Testbed for future works;

The MEC network testbed built for this work could apply to different platforms, software, and Radio Access Network Intelligent Controller (RIC) from Open Radio Access Network (O-RAN).

## 2. Network Architecture and Experiments Setup

The network architecture explanation consists of four subjects: 5G network schematic and dataflow path, network connection, MEC platform management, and MEC applications.

### 2.1. 5G Network Schematic and Dataflow Path

The 5G network schematic is based on the MEC initiative (Figure 1) and adds testing facilities such as a Vivotek camera (8 k high-definition), WiFi AP, laptop, Mi-Fi, and 4G/5G small cell, as shown in Figure 2 (left side). Moreover, the dataflow paths regarding different communication protocols were numbered from 1 to 4 in ellipse shape in Figure 2. The MEC platform utilizes the MEC solution developed by Chunghwa Telecom (CHT), the largest telecom company in Taiwan.

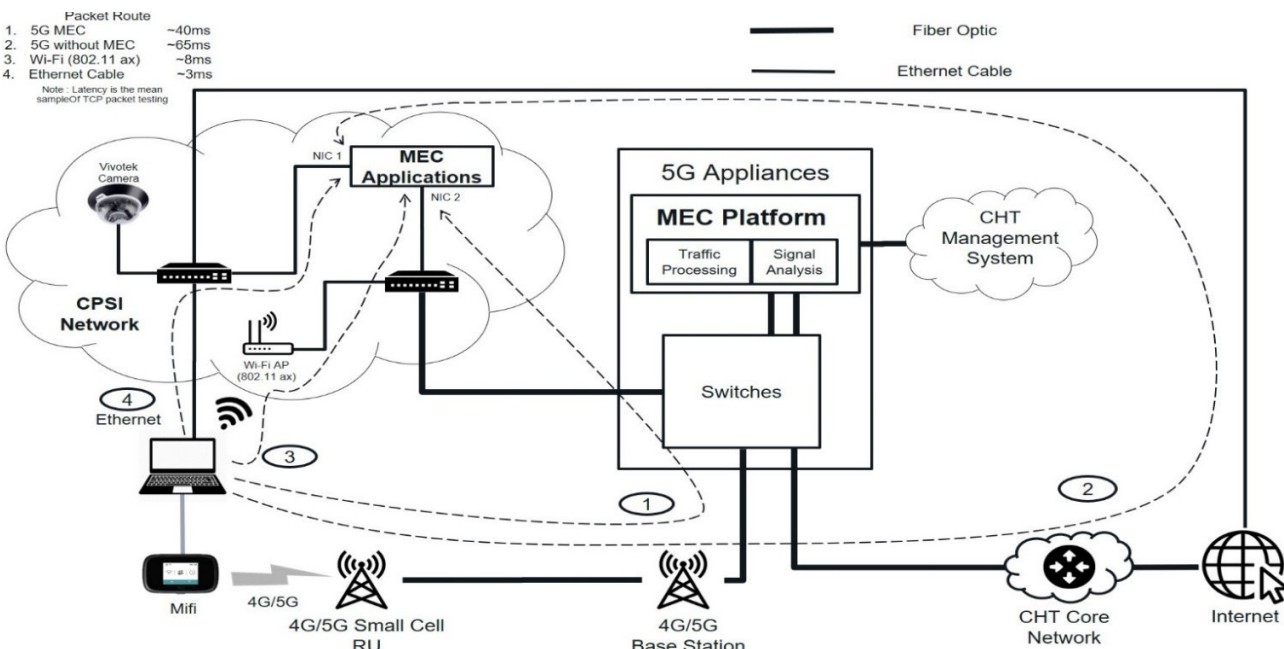

**Figure 2.** 5G network architecture and data transmission route: ①5G MES, ②5G without MEC, ③WiFi 6, (802.11ax) ④Ethernet.

4G/5G small cell radio unit (RU) is the access point of UE to the 5G MEC network, which is connected to the 4G/5G base station 400 m away via an optical fiber cord in the NTUST campus, as shown in Figure 2, lower left corner. Its transmission frequency band is in the 3500 MHz bands and is located on the workshop roof of the Industry 4.0 Implementation Center, approximately 4.5 m in height from the ground, for better signal reception quality.

This mobile network applies ENDC (E-UTRAN New Radio—Dual Connectivity) technology, which allows the UE to access the 5G and 4G LTE spectrum simultaneously (5G base station acting as the secondary node). The switches connected to the base station and backhaul regularly send a "heartbeat" to the MEC platform and other switches to detect updates and abnormalities. The switches send packets directly to the core network to maintain the regular operation of the base station and internet services for the user equipment.

The MEC platform provides a traffic offload function, analyzes users' internet traffic packets, and directs the corresponding packet to the MEC application. The MEC server, FWS-8600 (product of AAEON company), runs the MEC platform software.

### 2.2. Network Connection

There are two Network Interface Cards (NIC) on the server: one connects to the MEC platform, and the other connects to the campus network, the IP address assigned by the NTUST network. The setup allows the MEC applications to access the NTUST network in the local private network.

This scheme shows how to access the MEC applications deployed on the server via WiFi 6 (802.11ax) and Ethernet to campus networks at NTUST. Table 2 shows the specifications of the WiFi 6 (802.11ax) deployed on the network.

**Table 2.** WIFI 6 (802.11ax) specifications.

| Item | AX6000 WiFi Satellite (RBS850) Specifications [1] |
|---|---|
| WiFi Coverage | 2000 sq. ft |
| Orbi Satellite (AX6000) | Eight internal antennas with high-powered amplifiers each |
| AX6000 Tri-Band WiFi | 2.4 GHz (2400 Mbps), 5 GHz (2400 Mbps) |
| Satellite Ports | 4 Lan Gigabit Ethernet Ports |

[1] Product of NetGear: www.netgear.com (accessed on 3 March 2023).

### 2.3. MEC Platform Management

A dedicated line connects the MEC platform and the CHT management system to approach system information about the MEC server that runs the MEC platform. The management system could collect CPU, memory, and Network usage information. Besides collecting system information, the system allows CHT to manage the MEC server and platform outside the NTUST network domain.

### 2.4. MEC Applications

Figure 3 shows the components and applications on the MEC applications server. All the MEC application containers run on Docker for easy deployment and isolation. The Iperf3 server conducts performance testing to receive the generated packet and measure the network performance. The other container contains Nginx-RTMP Server, which can create a streaming service. The Nginx creates the RTMP server for Open Broadcaster Software to stream the Vivotek camera into the NGINX-RTMP server so that other users can access the live streaming from the Vivotek camera.

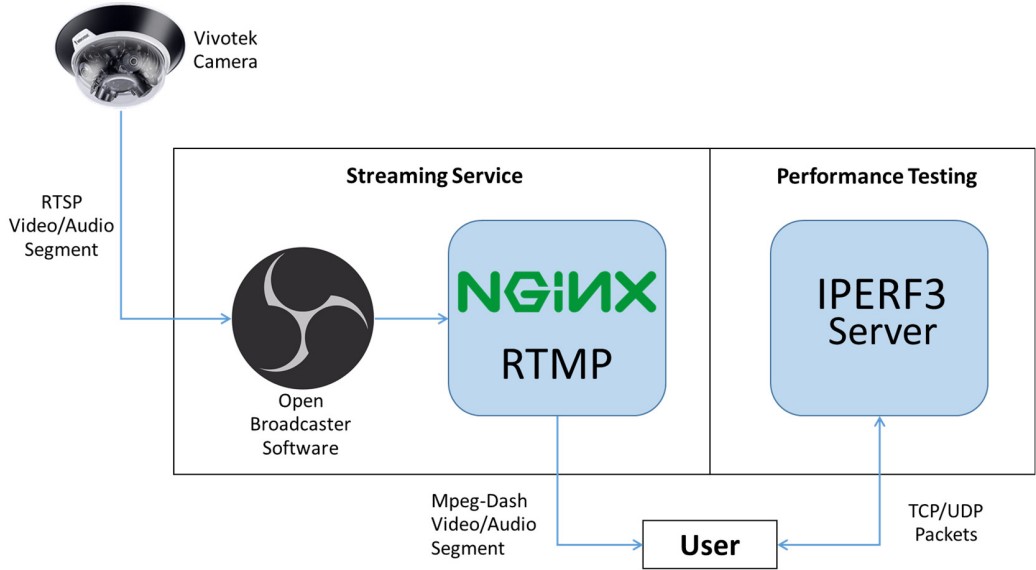

**Figure 3.** MEC applications—Communication model.

The communication models were different between Streaming Testing and Performance Testing. Table 3 shows the communication model between the user equipment and MEC applications. In that, the NGINX RTMP server performs the streaming testing via sent/received packets (video/audio), and the Iperf3 server conducts the performance testing by transmitting TCP/UDP packets.

**Table 3.** Communication Model.

| Testing | Packet Sent/Received | Applications |
|---|---|---|
| Streaming Testing | MPEG-DASH Video/Audio Segment | NGINX RTMP Server |
| Performance Testing | TCP/UDP Packets | Iperf3 Server |

### 2.5. Experiment Setup

Three laptops are engaged in the load test evaluation experiment to achieve up to 100 users in a virtual environment. Figure 4 illustrates the schematic of the #1 to #3 laptops connected to 5G MiFi and then wirelessly linked to 5G small cells for experiments—the 5G small cells set at the test site, Industry 4.0 Implementation Center, NTUST.

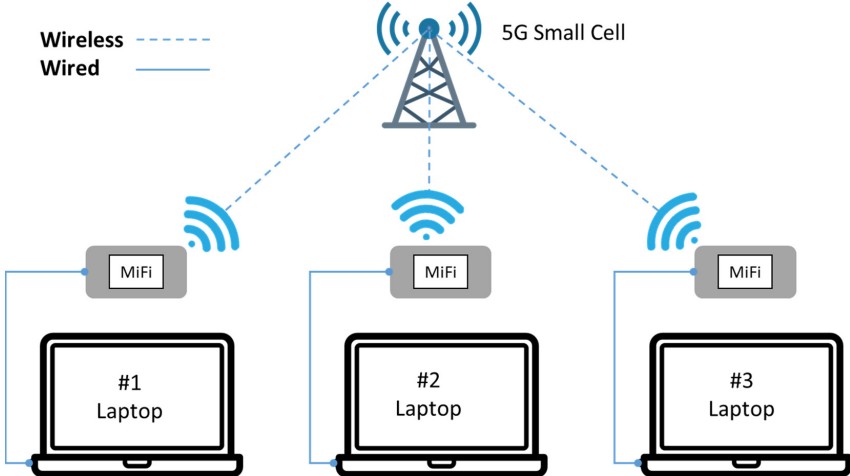

**Figure 4.** Schematic of the laptops, MiFi, and 5G small cell connection.

Figure 5a demonstrates the laptop and MiFi (5G router) setup for conducting load testing in various numbers of users. The MEC network performance and streaming test evaluation also use the same experimental setup. Figure 5b shows the network speed (Mb/s) and latency (ms) measurements on their displays regarding various parameters.

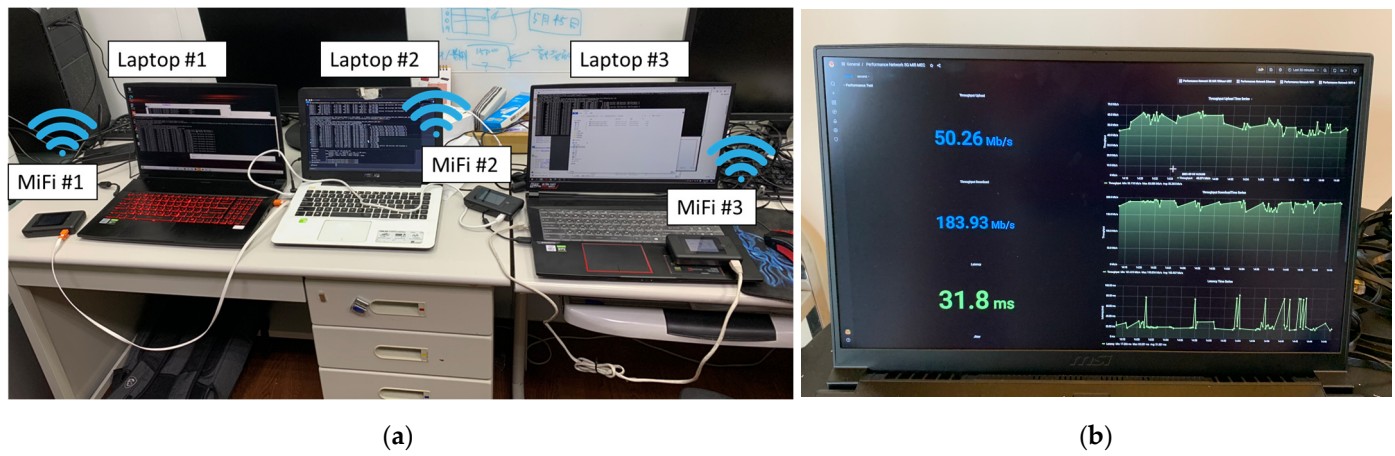

(**a**)                    (**b**)

**Figure 5.** Equipment setup and measurement in the display. (**a**) Laptop and MiFi (5G router) setup. (**b**) Network speed and latency measurement.

## 3. MEC Network Field Test

MEC network field test includes three portions: first, MEC network performance; second, streaming evaluation; and third, load test evaluation. These tests present the actual performance of the MEC network in the workshop with different configurations and the capability to handle streaming data and heavy data loading.

### 3.1. MEC Network Performance

A field test utilizes the actual equipment in the newly deployed 5G network but not a simulation, as Figure 2 shows. There are three kinds of access technologies to access the MEC network:

1.    An Ethernet cable connected to a switch to the NTUST campus network;
2.    WiFi 6 router connected to the NTUST network;
3.    5G Base Station ground to the MEC Server.

The Iperf3 tool applies to the network to measure the throughput, latency, jitter, and packet loss rate [14]. The Iperf3 python wrappers created the script for the client to generate TCP, ICMP, and UDP packets and a server to receive those packets. TCP packets test the

maximum available bandwidth, ICMP packets measure latency, and UDP packets count the jitter and loss packet rate.

The iperf3 client generates TCP/UDP packets, receives the packet, and generates the test result. TCP test conducts twenty parallel TCP flows to achieve the maximum throughput. For the UDP test, one UDP stream with a 10 Mbps bitrate maintains UDP windows size at 208 Kbytes and UDP packet size at 1470 bytes.

The Iperf3 client receives the test result data and saves the test result on a MySQL Server. The connection between the iperf3 client and server applied the WiFi 6 (802.11ax) and Ethernet. A 5G base station connects to the iperf3 server, and a 5G MiFi or a 5G mobile hotspot is used during the performance evaluation. In summary, there are four routes in this test:

- 5G (3.5 GHz) and 4G carriers connect the internet by passing through MEC;
- 5G (3.5 GHz) and 4G carriers connect the internet without passing through MEC;
- The Ethernet cable connects straight to the NTUST gateway (campus network);
- An 802.11ax WiFi (2.4 GHz 5 GHz) connection to a Wireless router goes to the NTUST gateway via an Ethernet cable.

### 3.2. Streaming Evaluation

Streaming evaluation is an essential index for measuring network performance because 95% of the global internet population watch YouTube and stream programs from there [15]. Moreover, the Cisco visual networking index shows video viewing can be up to 82% of internet traffic, and 22% of global IP could be ultra-high-definition video traffic by 2022 [16]. Regarding the above, the MEC application conducts video streaming tests to see the performance of a streaming service. The video source is $2688 \times 1920$ RTSP acquired from a Vivotek camera, transcoding into a $1280 \times 720$ (720P) MPEG-Dash video on the MEC application server. The Vivotek camera hardware specifications refer to Table 4.

The camera source utilizes the VIVOTEK MA9322 in the video streaming test. It acquired the video in 30FPS ($2688 \times 1920$) quality and applied the H.265 compression technology to achieve ultra-high quality videos reducing bandwidth in network traffic.

**Table 4.** VIVOTEK camera specifications.

| Nomination | MA9322-EHTV Specifications |
| --- | --- |
| CPU | Multimedia SoC (System-on-Chip) |
| RAM | 4 GB |
| Max Resolution | $2688 \times 1920$ [5MP] |
| Maximum Frame rate | 30 fps |
| Audio Capability | Two-Way Audio (Full Duplex) |
| Audio Interface | Built-in microphone |
| Network Interface | External microphone input External line output |
| Compression Technology | H.265 |

The test assumed that the Edge server has unlimited storage/cache, meaning all data could be received directly from the server using the network routes depicted in Figure 2.

The Jmeter [17], a load-testing tool combined with a streaming plugin, was used to test the streaming service's performance in the network. The plugin allows Jmeter to recognize and stream the MPEG-DASH video/audio segments on the MEC Applications server. Besides streaming the video, it collected data for analyzing the performance of the streaming service, such as data latency (time taken to send the first byte until the first byte) and response time (time taken to send the first byte until the last byte). The InfluxDB acts as a database to store all collected data and show the information by Grafana. Docker is the platform to deploy both InfluxDB and Grafana in the experiments.

### 3.3. Load Test Evaluation

A load test can examine the capacity or limitation of the MEC network. Load testing puts a system under extreme conditions to know the capacity or the limitation of a network

or system [18]. One of the stress testing conditions and scenarios is putting the system under normal or heavy load, referring to the works [19,20]. The scenarios and user numbers for this load test evaluation are as follows:

- 5 virtual users;
- 10 virtual users;
- 20 virtual users;
- 50 virtual users.

The heavy load tests create hundreds of virtual users to stream the video service on the MEC network. The Jmeter, an open-source testing tool developed by Apache Software Foundation (ASF), applied the same routes and access technology as the previous test. Three physical devices (laptops) create virtual users to conduct the network traffic. The data collected from the management system include network usage (Mbps), total memory usage (Gbyte), and total CPU usage (%).

## 4. MEC Network Test Results

This section presents the measurement data regarding MEC network performance, Streaming, and Load Test Evaluation. The measure includes data throughput, latency, jitter, the packet loss rate on network performance evaluation, hop and response time, streaming, and audio segment latency presented on streaming and load test evaluation.

### 4.1. MEC Network Performance

The performance evaluation test results are presented in Figures 6–9 regarding network throughput, latency, jitter, and packet loss rate. The network throughput with the different routes and access technology is present in the charts.

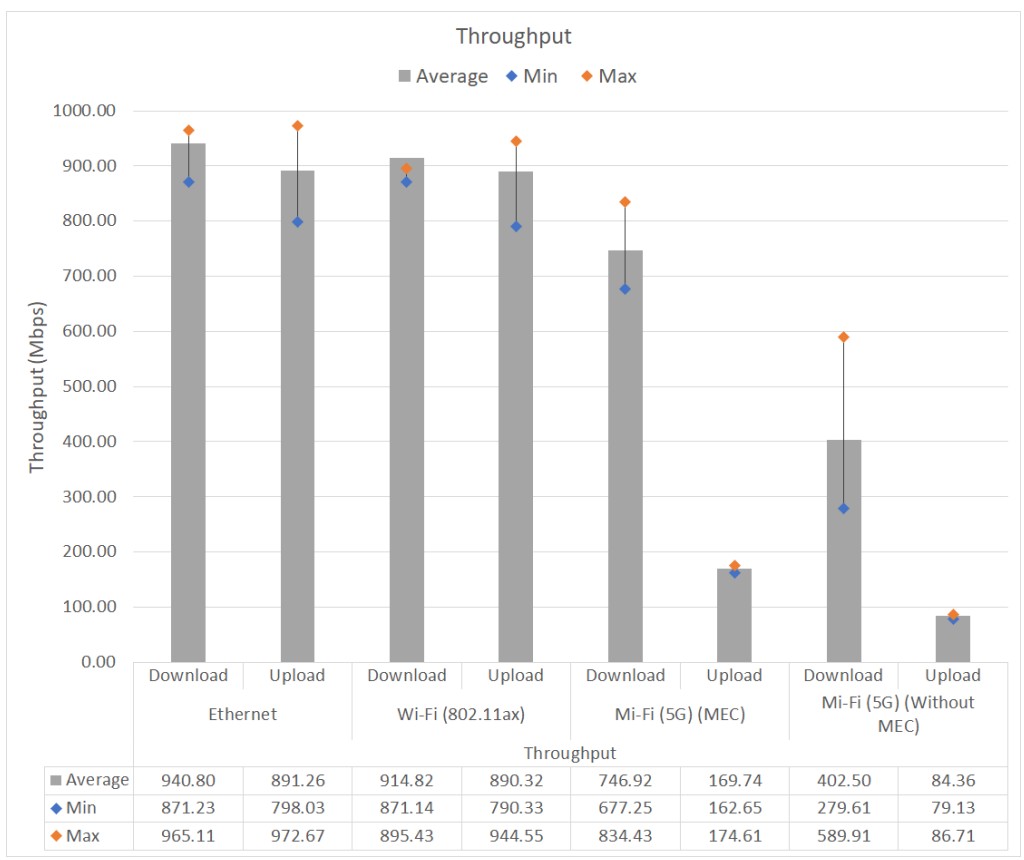

**Figure 6.** Multi-Access Edge Network Throughput.

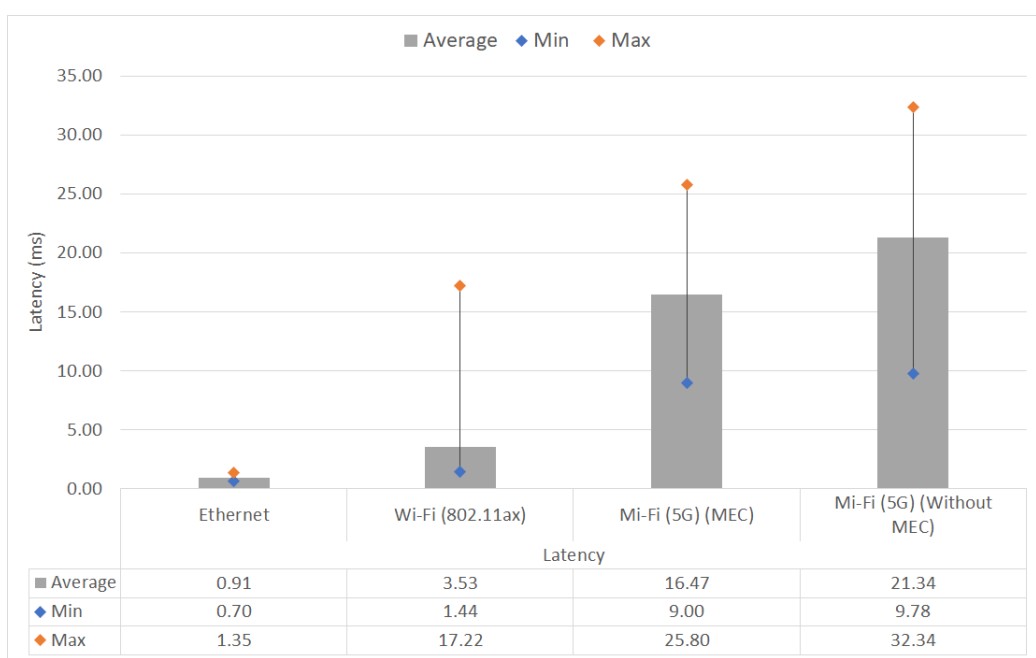

**Figure 7.** Multi-Access Edge Network Latency.

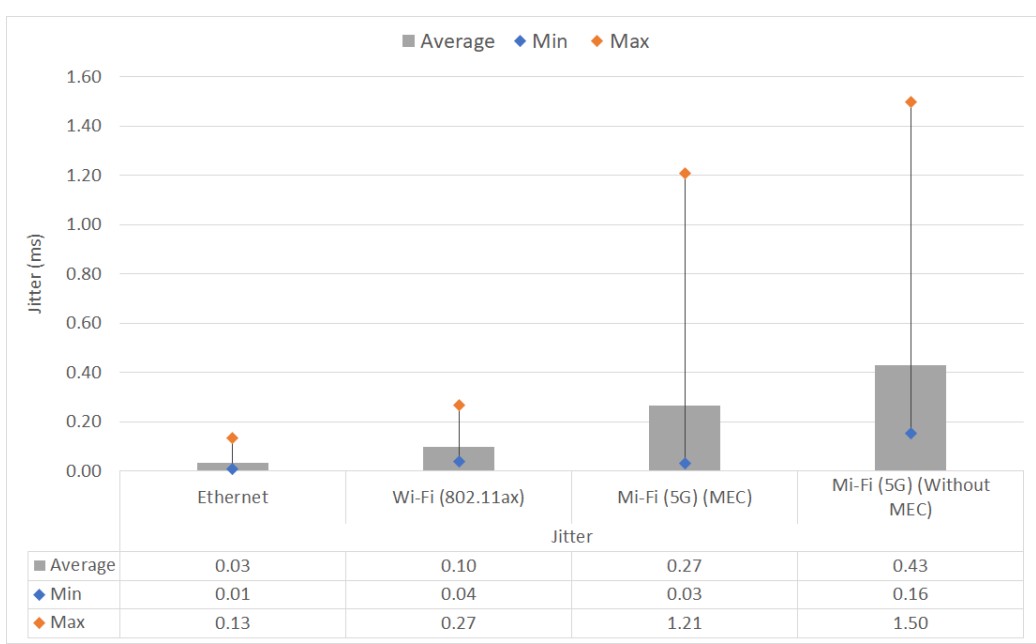

**Figure 8.** Multi-Access Edge Network Jitter.

The Iperf3 python wrappers created the script for the client to generate TCP, ICMP, and UDP packets and a server to receive those packets. This method can calculate the latency, jitter, and time consumption in receiving and processing the sent packets between the user and applications. TCP packets test the maximum available bandwidth, ICMP packets measure latency, and UDP packets count the jitter and loss packet rate.

Regarding the 5G MEC subject, Figure 6 shows that the 5G MEC has an average downlink throughput of above 746 Mbps and high up to 830 Mbps. Moreover, an average latency of 16 ms is presented in Figure 7, which could go as low as 9 ms. Further, Figure 8 depicts an average jitter of 0.27ms that could go low as 0.03 ms, and Figure 9 charts the packet loss of 0%.

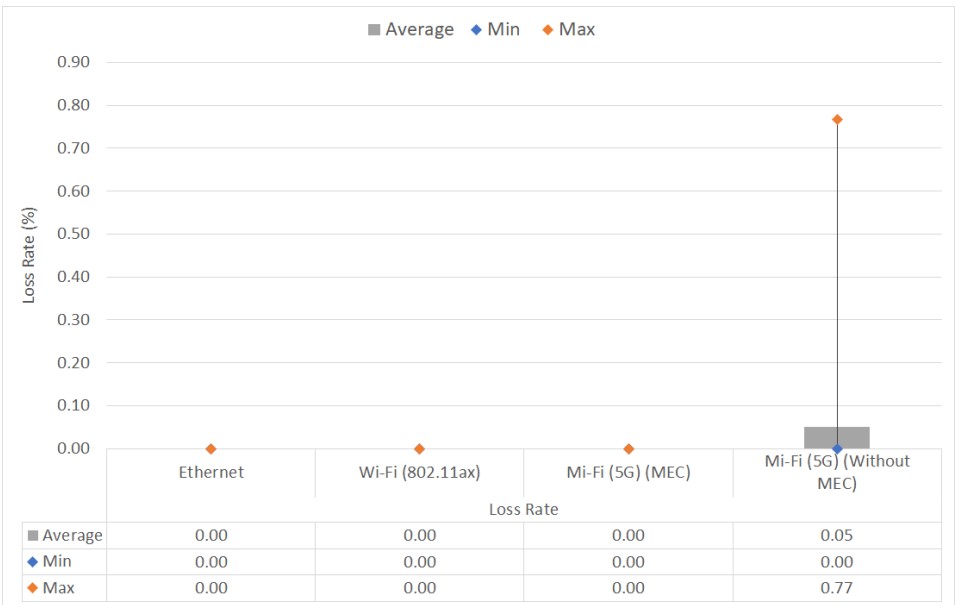

**Figure 9.** Multi-Access Edge Network Packet Loss Rate.

The MEC technology could increase the maximum throughput and lower the average jitter, latency, and loss rate by shortening the route from the user to the application destination. The results can meet the requirement of the Intelligent Transport System and Smart Grid, which requires a bandwidth of 10–700 Mbps and latency of below 20 ms [21].

The WiFi 6 (802.11ax) route performed a downlink throughput of 914 Mbps and an uplink throughput of 890 Mbps. Even though the WiFi 6 datasheet shows its capacity of up to 2.4 Gbps, the maximum measured speed is only 914 Mbps because of the constraints on the 1 Gbps switch. Comparing 5G MEC to the other network performances besides Ethernet, 5G MEC only has better throughput than 5G without MEC. 5G MEC latency and jitter are higher than WiFi 6 (802.11ax) because our WiFi 6 (802.11ax) system has a shorter route and hop than 5G MEC.

### 4.2. Streaming Test Evaluation

Figures 10 and 11 show the streaming performance in latency and response time between different access technologies on the MEC network and WiFi 6.

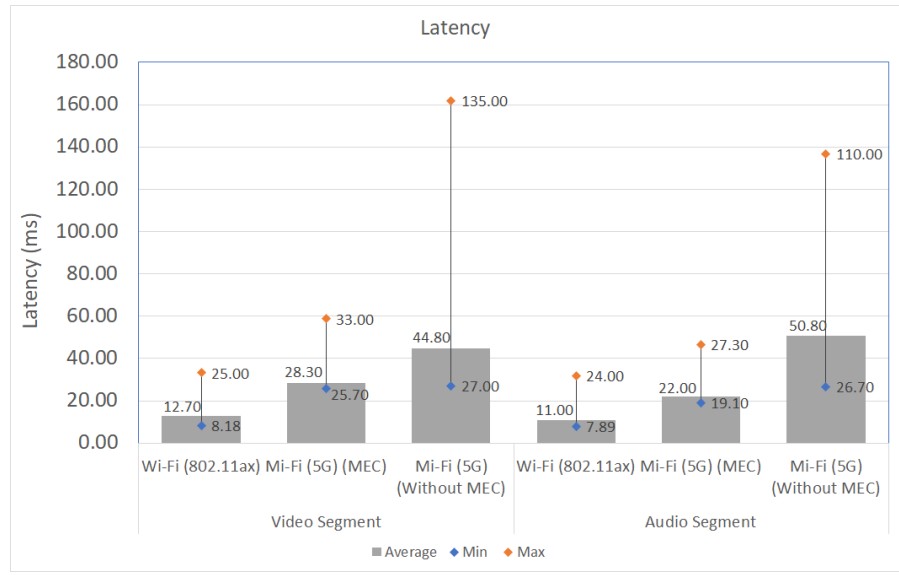

**Figure 10.** Schemes follow the same formatting.

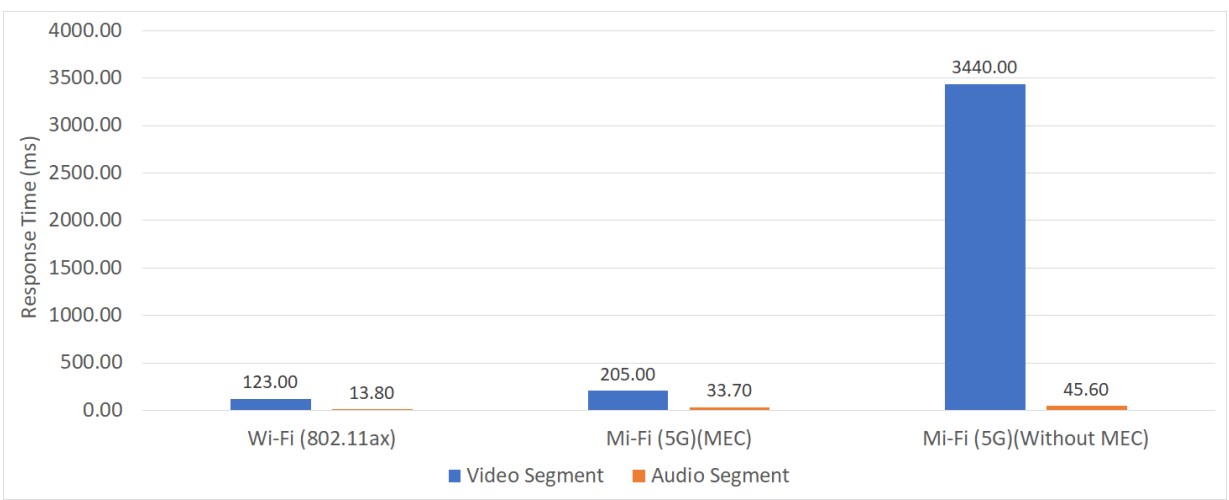

**Figure 11.** Response Time in Streaming Test Result.

5G MEC performs better than 5G MEC without MEC when streaming a 720p MPEG-Dash video. 5G MEC has a 28 ms average latency for the video segment and 22 ms for the audio segment. The latency for both video and audio on the 5G MEC never goes above 33 ms, which means that 5G MEC could perform an ultra-low latency streaming service below 50 ms. Even though 5G without MEC has an average latency below 50 ms, the instability of 5G without MEC, which has a maximum latency of 135 ms, could not replicate an ultra-low latency streaming service such as 5G MEC. Of the results that show the response time of both 5G MEC and without MEC, 5G MEC has an average response time of 205 ms, while 5G without MEC has an average response time of 3–4 s. Compared to WiFi 6 (802.11ax), 5G MEC has a higher latency in the streaming service because the WiFi 6 (802.11ax) has a shorter route on the network. This result proves that MEC technology is improving the service quality of the experience by shortening the route from the user to the application and reducing latency.

The 5G without MEC route response time is considerable compared to 5G with MEC because of the number of hops that 5G without MEC has to jump through. Based on the number of hops and the response time from all hops, it could be seen that some hops have a longer response time, becoming the bottleneck on the 5G without the MEC route in this streaming case. Since the hop on 5G MEC and WiFi 6 is very short, the response time becomes very low in this scenario, as Figure 11 shows. The response time of WiFi 6 can even be shorter than the 5G MEC route, according to the charts in Figure 11, left and center.

*4.3. Load Test Evaluation*

The stress test results present the streaming service capability of the 5G MEC network. 5G MEC maximum users in 5, 10, 20, 50, and 100 can achieve ultra-low latency below 50 ms, 70 ms, 350 ms, 500 ms, and 1 s, respectively, as shown in Figures 12 and 13.

The MEC management system, AAEON MEC server, shows usage of 10% CPU and 6.17% memory while handling 340 Mbps traffic under 100 virtual users during the test. Table 5 shows the data acquired from the management system during the load test on the MEC network.

The WiFi 6 (802.11ax) performs better than 5G MEC when virtual users have more than 10 users. Below 10 virtual users, WiFi 6 (802.11ax) has a lower average latency due to the shorter route, but when the number of virtual users increases, WiFi 6 (802.11ax) average latency is higher than 5G MEC. This result means that 5G MEC handles massive traffic and users better than Wi-Fi6(802.11ax). That could also mean our 5G MEC system has better performance handling requests and more extensive traffic than our WiFi 6 (802.11ax) system.

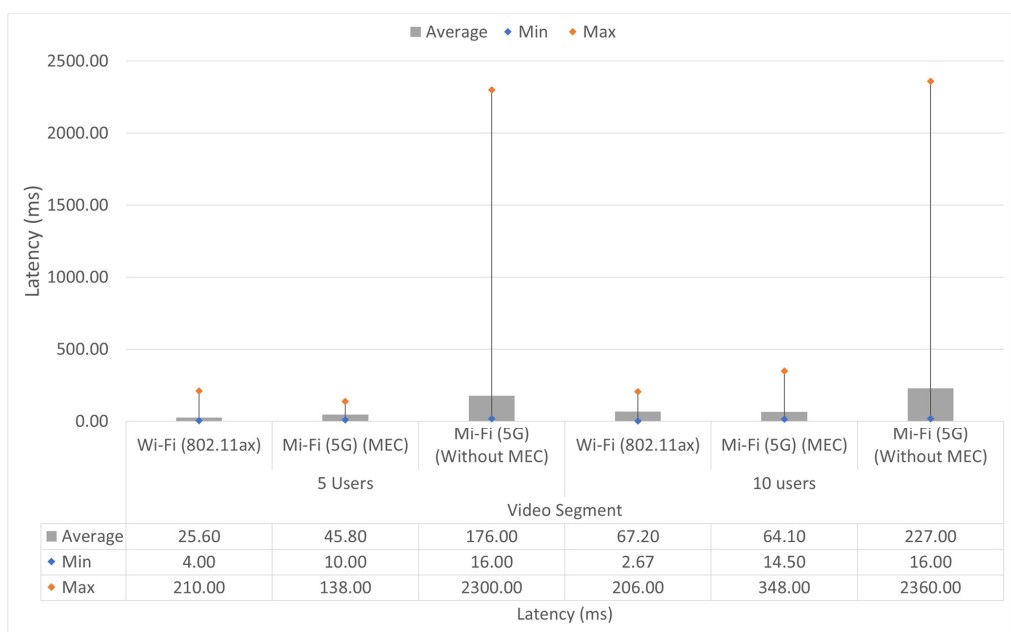

**Figure 12.** Load Test 5 & 10 Users Latency Result.

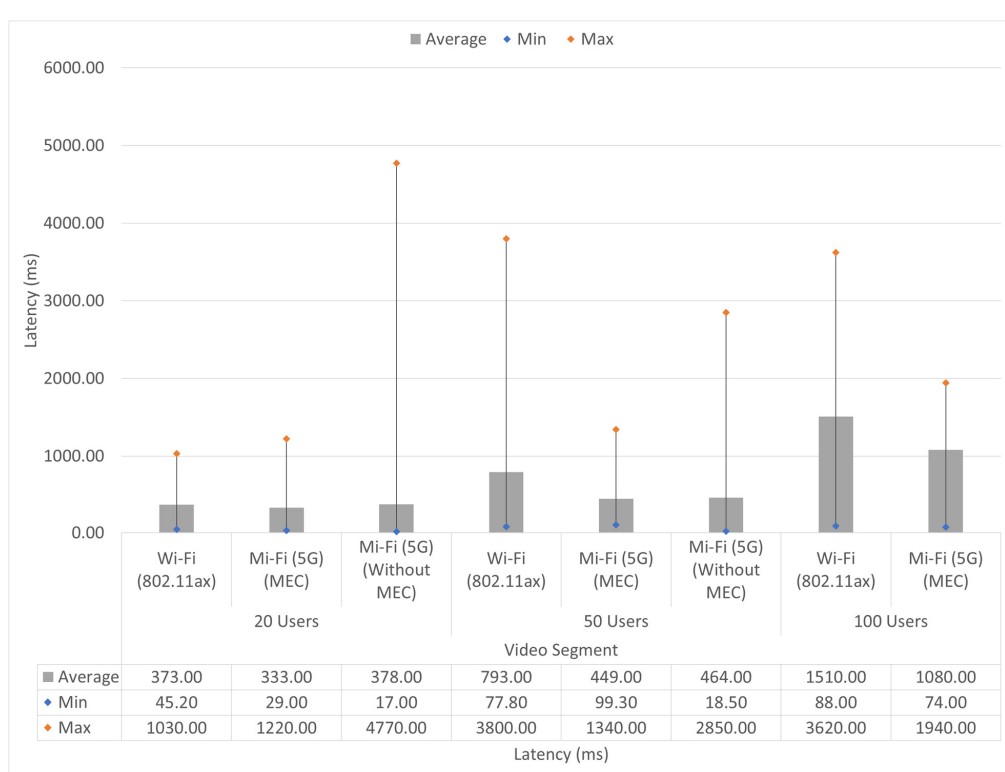

**Figure 13.** Load Test 20, 50, & 100 Users Latency Result.

**Table 5.** Load test MEC management system result.

| Users | Total Traffic (Mbps) | Packet Loss Rate (%) | CPU Usage (%) | Memory Usage (%) |
|---|---|---|---|---|
| 5 | 77.73 | 0 | 10 | 6.15 |
| 10 | 160.64 | 0 | 10 | 6.15 |
| 20 | 330.64 | 0 | 10 | 6.15 |
| 50 | 331.22 | 0 | 10 | 6.16 |
| 100 | 344.32 | 0 | 10 | 6.17 |

## 5. Conclusions and Future Work

This article presents the network performances of different access technologies and routes. Through testing for proof of concept, which uses 5G radio and MEC technology to shorten the route between user and application on the network, we show that 5G MEC has a better network performance than 5G without MEC. We also have the result that shows 5G MEC gives us improvement in services such as streaming video services. Finally, we show that 5G MEC performs better in handling traffic and requests created by 10 virtual users and above on the network than our WiFi 6 (802.11ax) and 5G without MEC network. We conclude that 5G MEC is an essential technology for better network performances for our applications development that we want to have on our building and network. In future works, we should eliminate the physical bottleneck, such as the slow 1 Gbps switch, and acquire the authorization on the SDN and NFV for better control and setup in the network.

In this work, we demonstrated the performance of 5G MEC using real devices, networks, and programs to discover the network speed and latency instead of simulations. In the future, we can utilize the achievement and the test bed built to extend the research into 5G Core technology such as SDN, NFV, network slicing, and 5G relevant applications.

**Author Contributions:** Conceptualization, R.-G.C. and N.M.R.; methodology, R.-G.C. and N.M.R.; software, N.M.R. and C.-W.T.; validation, N.M.R. and C.-W.T.; resources, Y.-F.L., W.-H.Y.; writing—original draft preparation, N.M.R. and S.-H.L.; writing—review and editing, J.-T.W. and S.-H.L.; supervision, R.-G.C.; project administration, R.-G.C.; funding acquisition, R.-G.C. All authors have read and agreed to the published version of the manuscript.

**Funding:** This research was funded by 111-2218-E-011-014, the funded project of the National Science and Technology Council (NSTC), Taiwan.

**Institutional Review Board Statement:** Not applicable.

**Informed Consent Statement:** Not applicable.

**Data Availability Statement:** The data presented in this study are available on request from the corresponding author. The data are not publicly available due to their use in future research activities.

**Acknowledgments:** We appreciated the technical team providing a self-developed MEC solution, setting up 5G network facilities, and technical support from Chunghwa Telecom, the largest Mobile Network Operator telecom company in Taiwan.

**Conflicts of Interest:** The authors declare no conflict of interest.

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
