# Peer review of "Implementation and Evaluation of 5G MEC-Enabled Smart Factory"

_electronics, doi:10.3390/electronics12061310_

Round 1

Reviewer 1 Report

In this paper, the authors study the performance of 5G network with and without MEC, while comparing the results to various wired and wireless networks. The paper is well structured and divided in equitable sections that makes it easy to follow and to understand. The simulation is well described and the obtained results seems convincing. However, I have the following comments before any possible publication of the manuscript:

1- Try to summarize the entire obtained results in the abstract, not only the latency metric.

2- The literature review of the manuscript should be given more attention. Please mention more recent references similar to yours. Then, build a table to show the differences between the state of the art and your work.

3- In the results section, the authors mention what they observe in the figures without explaining why we arrived to the results.

4- Please justify the selection of Vivotek camera and if the obtained results can be generalized to any other types.

Author Response

Thank you for your previous comments and suggestions regarding the quality of our manuscript. The resubmitted version (.doc) keeps track-change and markups for easy identifying the revised contents. Please turn on Track Changes and show the markups in Word when reviewing.  

Reviewer 2 Report

Dear Authors,

I want to have clarity on the below mentioned point as i could not find these in your manuscript.

1) clarity between Network model and communication model

2) how you are calculating (i) time consumed when the user offloads the task, (ii) time consumed when computation tasks are processed on the edge cloud i.e MEC

3) What about experiment set up ?

4) in results what about (i) Impact of computation amount on task duration. (ii) Impact of computation amount on energy cost.

5. Comparison With Other Methods 

Author Response

Thank you for your previous comments and suggestions regarding the quality of our manuscript. The resubmitted version (.doc) keeps the correction and notification for easy identifying the revised contents. Please turn on Track Changes and show the markups in Word when reviewing.     

Round 2

Reviewer 2 Report

All reviewers' comments have been addressed. The manuscript can now be accepted.